# An Exploration of a Balanced Up-Downwind Scheme for Solving Heston Volatility Model Equations on Variable Grids

**Chong Sun * and Qin Sheng**

Department of Mathematics and Center for Astrophysics, Space Physics, and Engineering Research, Baylor University, One Bear Place, Waco, TX 76798-7328, USA; qin_sheng@baylor.edu
* Correspondence: chong_sun@baylor.edu

**Abstract:** This paper studies an effective finite difference scheme for solving two-dimensional Heston stochastic volatility option-pricing model problems. A dynamically balanced up-downwind strategy for approximating the cross-derivative is implemented and analyzed. Semi-discretized and spatially nonuniform platforms are utilized. The numerical method comprised is simple and straightforward, with reliable first order overall approximations. The spectral norm is used throughout the investigation, and numerical stability is proven. Simulation experiments are given to illustrate our results.

**Keywords:** Heston volatility model; initial-boundary value problems; finite difference approximations; up-downwind scheme; order of convergence; stability

---

## 1. Introduction

Demand for highly effective, efficient, and reliable numerical methods has grown increasingly high for solving option-trading modeling equations involving cross-derivative terms. However, desirable computational procedures are, in general, difficult to obtain, due to challenges from the participation of cross-derivatives [1,2]. This motivates our study. In this investigation, targeted at European options that can only be exercised on dates of maturity, we propose and analyze a new and dynamically balanced up-downwind finite difference method in the pursuit.

In the early 1970s, Black, Scholes, and Merton introduced the popular Black-Scholes-Merton (BSM) model [3,4]. Under their consideration, stock prices were assumed to follow geometric Brownian motion, while the volatility of the stock prices was fixed and no sudden jumps occurred. However, classic BSM models often cannot fit ideally into the market data observed nowadays [4]. This may be due to the fact that, in modern financial markets, not only are stock prices subject to risk, but also the estimate of riskiness is typically subject to significant uncertainty. The Black-Scholes model does not adequately take into account essential characteristics of market dynamics, such as fat tails, skewness of the distribution of log returns, and the correlation between the value of the underlying and its volatility. It has also been observed that the volatility starts to fluctuate when the market reacts to new information [5]. To incorporate an additional source of randomness into an option pricing model, Heston proposed a more refined approach, based on the concept of stochastic volatility [5–7].

A closed-form solution of the model was also obtained by Heston, under a set of specific boundary and initial values for assets of the European type [6]. However, to meet a growing demand for American options and other assets, pricing equations often need to be placed together with more realistic initial boundary conditions, or even free boundary conditions. Closed forms of solutions are, in general, unavailable. Thus, numerical approximations of such solutions have become important and necessary.

This paper is concerned with European options. The scheme we developed here can be extended directly to price American options. However, due to the free boundary conditions associated with the American options, the stability and convergence analysis of the scheme becomes rather complicated. We are still working on the theoretical results behind applying our scheme to American options.

There have been numerous recent publications on the numerical solution of Heston modeling equations. For instance, certain first-order up-downwind algorithms were proposed and studied by Ma and Forsyth [2]. Stability analysis was carried out via standard von Neumann analysis for Cauchy problems or problems with periodic boundary conditions [8,9]. Although numerical stabilities have been under investigation for even high order schemes on nonuniform grids [4], rigorous analysis is only available in cases where the cross-derivative terms are neglected. The challenge for stability analysis persists, whenever a cross-derivative or more general boundary data structure exists [2,10].

However, cross-derivatives are essential to partial differential equations modeling a Heston Process. Further, Heston modeling formulations also require more realistic Dirichlet, Neumann, or mixed boundary conditions [6,11]. These have motivated our approaches. In this paper, we are particularly interested in computations based on a Heston put option model [4,8,9,12–14]. We are primarily interested in a linearly stable finite difference method, based on nonuniform grids. Our intention is to effectively reduce the computational costs and raise the algorithmic efficiency, by way of application of appropriate adaptive mechanisms. These results can be extended for call options, in similar ways.

In particular, we consider the following two-dimensional Heston volatility model interpreting the behavior of the asset value $S$ and its volatility $y$ at time $t \geq 0$,

$$\frac{dS(t)}{S(t)} = \mu dt + \sqrt{y(t)}dW_1(t), \tag{1}$$

$$dy(t) = \kappa(\eta - y(t))dt + \sigma\sqrt{y(t)}dW_2(t), \tag{2}$$

$$\text{cov}(dW_1(t), dW_2(t)) = \rho dt, \tag{3}$$

where $\mu$ is the expected return of the asset, $\kappa$ is the rate of reversion to the mean level of the volatility, $\eta$ is the mean level of the volatility, $\sigma > 0$ is the volatility parameter, and $\text{cov}(u, v)$ is the covariance between $u$ and $v$ [6,15]. The two Wiener processes $W_1(t)$ and $W_2(t)$ describe the random noise in asset and volatility, respectively; they are assumed to be correlated with a constant correlation coefficient $\rho \in [-1, 1]$.

Let $v(S, y, t)$, $t \geq 0$, denote the value of a European put option that is a function of asset price, volatility, and time. An application of Itô's Lemma and the non-arbitrage principle with a riskless portfolio construction leads to [4,6,9,16,17],

$$v_t + \frac{1}{2}yS^2v_{SS} + \rho\sigma ySv_{Sy} + \frac{\sigma^2 y}{2}v_{yy} + rSv_S + \kappa(\eta - y)v_y = rv, \quad S, y > 0. \tag{4}$$

Let

$$v(S, y, T) = \max\{K - S, 0\}, \quad S, y \geq 0,$$

be the terminal condition to use, where $T$ is the payoff time and $K$ is the strike price. We adopt the following mixed boundary conditions for $S, y > 0$ and $T > t \geq 0$ [8]:

$$v(0, y, t) = Ke^{-r(T-t)}, \tag{5}$$

$$\lim_{S \to \infty} v(S, y, t) = 0, \tag{6}$$

$$v_y(S, 0, t) = 0, \tag{7}$$

$$\lim_{y \to \infty} v_y(S, y, t) = 0. \tag{8}$$

Set $\tau = T - t$. Equation (4) can be rewritten as

$$v_\tau = \frac{yS^2}{2}v_{SS} + \rho\sigma y S v_{Sy} + \frac{\sigma^2 y}{2}v_{yy} + rSv_S + \kappa(\eta - y)v_y - rv, \quad T > \tau > 0.$$

Let $x = \ln\frac{S}{K}$ and $u = \frac{v}{K}e^{r\tau}$. For $-\infty < x < \infty$, $y > 0$, $T > \tau > 0$, we observe that

$$u_\tau = \frac{y}{2}u_{xx} + \rho\sigma y u_{xy} + \frac{\sigma^2 y}{2}u_{yy} - \left(\frac{y}{2} - r\right)u_x + \kappa(\eta - y)u_y, \tag{9}$$

together with constraints [4,8,14],

$$
\begin{align}
u(x, y, 0) &= \max\{1 - e^x, 0\}, & -\infty < x < \infty,\ y > 0, \tag{10}\\
\lim_{x \to -\infty} u(x, y, \tau) &= 1, & y > 0,\ T \geq \tau > 0, \tag{11}\\
\lim_{x \to \infty} u(x, y, \tau) &= 0, & y > 0,\ T \geq \tau > 0, \tag{12}\\
u_y(x, 0, \tau) &= 0, & -\infty < x < \infty,\ T \geq \tau > 0, \tag{13}\\
\lim_{y \to \infty} u_y(x, y, \tau) &= 0, & -\infty < x < \infty,\ T \geq \tau > 0. \tag{14}
\end{align}
$$

We may extend the temporal domain for (9)–(14) by allowing $T = \infty$. Further, for the sake of computation, we consider a truncated spatial domain $\Omega = \{(x, y) : -X < x < X;\ 0 < y < Y\}$, for sufficiently large $X$ and $Y$, in the rest of our investigation.

In the next section, a nonuniform spatial mesh will be introduced. Based on it, a semi-discretized system will be derived for solving (9)–(14). Dynamically balanced up-downwind difference approximations will be presented. A general linear stability analysis and computational experiments will be carried out in Section 2. Computationally evaluated rates of convergence of the scheme will also be provided. Finally, conclusions and future research intentions will be given in Section 3.

## 2. Results

### 2.1. Balanced Up-Downwind Semi-Discretized Scheme

Let $-X = x_0 < x_1 < \cdots < x_M < x_{M+1} = X$, $0 = y_0 < y_1 < \cdots < y_N < y_{N+1} = Y$, for which $x_m - x_{m-1} = h_m$, $y_n - y_{n-1} = k_n$, $0 < h_m, k_n \ll 1$, $m = 1, 2, \ldots, M + 1$, $n = 1, 2, \ldots, N + 1$.

Let $z_{m,n} = z_{m,n}(\tau)$ be an approximation of $z(x_m, y_n, \tau)$, $0 \leq m \leq M + 1$, $0 \leq n \leq N + 1$, $0 < \tau < T$. Further, let $\Delta_{\ell,+}$, $\Delta_{\ell,-}$, and $\Delta_{\ell,0}$ be forward, backward, and central difference operators in the $\ell$-direction, respectively, where $\ell \in \{x, y\}$ [10,18]. Similarly, for appropriate indexes, we define

$$\Delta_{x,0}^2 z_{m,n} = \frac{2z_{m+1,n}}{h_{m+1}(h_{m+1} + h_m)} - \frac{2z_{m,n}}{h_{m+1}h_m} + \frac{2z_{m-1,n}}{h_m(h_{m+1} + h_m)}, \tag{15}$$

$$\Delta_{y,0}^2 z_{m,n} = \frac{2z_{m,n+1}}{k_{n+1}(k_{n+1} + k_n)} - \frac{2z_{m,n}}{k_{n+1}k_n} + \frac{2z_{m,n-1}}{k_n(k_{n+1} + k_n)}. \tag{16}$$

We now approximate the diffusion terms in (9) by using the above, and the derivatives in (13) and (14) via the following,

$$u_y(x_m, 0, \tau) \approx \frac{1}{h_y}\Delta_{y,+}u_{m,0}(\tau),\ u_y(x_m, Y, \tau) \approx \frac{1}{h_y}\Delta_{y,-}u_{m,N+1}(\tau),\ 0 < \tau < T.$$

We approximate the advection terms in (9) through three different channels, depending upon relation between $\eta$ and $r$.

Case 1: $\eta > 2r$.

$$u_x(x_m, y_n, \tau) \approx \Delta_{x,+}u_{m,n}, \ u_y(x_m, y_n, \tau) \approx \Delta_{y,+}u_{m,n}, \qquad 2r \geq y > 0, \qquad (17)$$

$$u_x(x_m, y_n, \tau) \approx \Delta_{x,-}u_{m,n}, \ u_y(x_m, y_n, \tau) \approx \Delta_{y,+}u_{m,n}, \qquad \eta \geq y > r, \qquad (18)$$

$$u_x(x_m, y_n, \tau) \approx \Delta_{x,-}u_{m,n}, \ u_y(x_m, y_n, \tau) \approx \Delta_{y,-}u_{m,n}, \qquad Y > y > \eta. \qquad (19)$$

Case 2: $\eta \leq 2r$.

$$u_x(x_m, y_n, \tau) \approx \Delta_{x,+}u_{m,n}, \ u_y(x_m, y_n, \tau) \approx \Delta_{y,+}u_{m,n}, \qquad \eta \geq y > 0, \qquad (20)$$

$$u_x(x_m, y_n, \tau) \approx \Delta_{x,+}u_{m,n}, \ u_y(x_m, y_n, \tau) \approx \Delta_{y,-}u_{m,n}, \qquad 2r \geq y > \eta, \qquad (21)$$

$$u_x(x_m, y_n, \tau) \approx \Delta_{x,-}u_{m,n}, \ u_y(x_m, y_n, \tau) \approx \Delta_{y,-}u_{m,n}, \qquad Y > y > 2r. \qquad (22)$$

The computational stencils for the Case 1 and Case 2 are given in Figure 1.

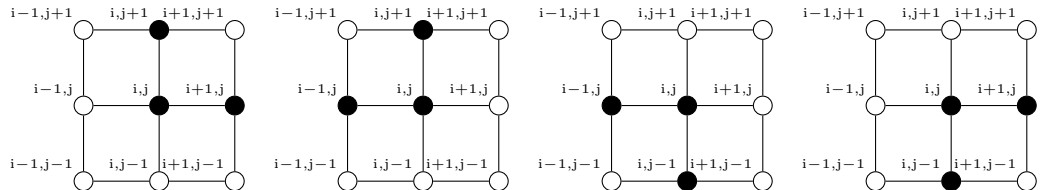

**Figure 1.** Computational stencil for (17) and (20) [frame 1]; (18) [frame 2]; (19) and (22) [frame 3]; and (21) [frame 4].

Define

$$h_{\min} = \min_{m=1,2\cdots M} h_m, \ h_{\max} = \max_{m=1,2\cdots M} h_m; \ k_{\min} = \min_{n=1,2\cdots N} k_n, \ k_{\max} = \max_{n=1,2\cdots N} k_n.$$

We now approximate the cross-derivative in (9) dynamically. To this end, we have the following two cases presented in Sections 2.1.1 and 2.1.2.

2.1.1. Case for $\rho \in [-1, 0]$

For smoothness of the nonuniform grids [18], we require that

$$-\rho k_{\max} \leq \sigma h_{\min} \leq \sigma h_{\max} \leq -\frac{1}{\rho}k_{\min}. \qquad (23)$$

We propose that

$$u_{xy}(x_m, y_n, \tau) = \frac{1}{2}(\Delta_{x,+}\Delta_{y,-} + \Delta_{x,-}\Delta_{y,+})u_{m,n}(\tau) + \mathcal{O}(h_{\max} + k_{\max}). \qquad (24)$$

Substitute all spatial derivative approximations into (9), and let $w$ denote the approximate solution to $u$. We acquire the following linear system,

$$w'(\tau) = Aw(\tau) + f(\tau), \qquad (25)$$

where $w, f \in \mathbb{R}^{MN}$ and $A \in \mathbb{R}^{MN \times MN}$ is block tridiagonal of the form

$$
A \;=\; \begin{bmatrix}
D_1 & Q_1 & \cdots & \cdots & \cdots & 0 \\
P_2 & D_2 & Q_2 & \cdots & \cdots & 0 \\
\vdots & \ddots & \ddots & \ddots & \cdots & \vdots \\
\vdots & \cdots & P_{M-2} & D_{M-2} & Q_{M-2} & 0 \\
\cdots & \cdots & \cdots & P_{M-1} & D_{M-1} & Q_{M-1} \\
0 & \cdots & \cdots & \cdots & P_M & D_M
\end{bmatrix},
$$

where $P_i, D_j, Q_k \in \mathbb{R}^{N \times N}$, $i = 2, 3, \ldots, M$; $j = 1, 2, \ldots, M$; $k = 1, 2, \ldots, M-1$. Nontrivial entries of the matrices $P_m$, $D_m$, and $Q_m$, for their respective ranges of $m$, are as follows:

$$
p_{n,n}^{(m)} \;=\; \begin{cases}
\dfrac{y_n}{h_m(h_m + h_{m+1})} + \dfrac{\rho\sigma y_n}{2h_m k_{n+1}}, & 0 < y_n \le 2r, \\[3mm]
\dfrac{y_n}{h_m(h_m + h_{m+1})} + \dfrac{\rho\sigma y_n}{2h_m k_{n+1}} + \dfrac{y_n - 2r}{2h_m}, & 2r < y_n < Y - k_{N+1}, \\[3mm]
\dfrac{y_N}{h_m(h_m + h_{m+1})} + \dfrac{y_N - 2r}{2h_m}, & y_n = Y - k_{N+1};
\end{cases}
$$

$$
p_{n,n+1}^{(m)} \;=\; -\dfrac{\rho\sigma y_n}{2h_m k_n};
$$

$$
d_{n,n-1}^{(m)} \;=\; \begin{cases}
\dfrac{\sigma^2 y_n}{k_n(k_n + k_{n+1})} + \dfrac{\rho\sigma y_n}{2h_{m+1} k_n}, & k_1 < y_n \le \eta, \\[3mm]
\dfrac{\sigma^2 y_n}{k_n(k_n + k_{n+1})} + \dfrac{\rho\sigma y_n}{2h_{m+1} k_n} - \dfrac{\kappa(\eta - y_n)}{k_n}, & \eta < y_n \le Y - k_{N+1};
\end{cases}
$$

$$
d_{n,n}^{(m)} \;=\; \begin{cases}
\alpha_{m,1} + \dfrac{y_1 - 2r}{2h_{m+1}} - \dfrac{\kappa(\eta - y_1)}{k_2}, & y_n = k_1, \\[3mm]
\beta_{m,n} + \dfrac{y_n - 2r}{2h_{m+1}} - \dfrac{\kappa(\eta - y_n)}{k_{n+1}}, & k_1 < y_n \le 2r, \\[3mm]
\beta_{m,n} - \dfrac{y_n - 2r}{2h_m} - \dfrac{\kappa(\eta - y_n)}{k_{n+1}}, & 2r < y_n \le \eta, \\[3mm]
\beta_{m,n} - \dfrac{y_n - 2r}{2h_m} + \dfrac{\kappa(\eta - y_n)}{k_n}, & \eta < y_n < Y - k_{N+1}, \\[3mm]
\gamma_{m,N} - \dfrac{y_N - 2r}{2h_m} + \dfrac{\kappa(\eta - y_N)}{k_N}, & y_N = Y - k_{N+1};
\end{cases}
$$

$$
d_{n,n+1}^{(m)} \;=\; \begin{cases}
\dfrac{\sigma^2 y_n}{k_{n+1}(k_n + k_{n+1})} + \dfrac{\rho\sigma y_n}{2h_m k_{n+1}} + \dfrac{\kappa(\eta - y_n)}{k_{n+1}}, & 0 < y_n \le \eta, \\[3mm]
\dfrac{\sigma^2 y_n}{k_{n+1}(k_n + k_{n+1})} + \dfrac{\rho\sigma y_n}{2h_m k_{n+1}}, & \eta < y_n < Y - k_{N+1};
\end{cases}
$$

$$
q_{n,n-1}^{(m)} \;=\; -\dfrac{\rho\sigma y_n}{2h_{m+1} k_n}, \quad y_n > k_1;
$$

$$
q_{n,n}^{(m)} \;=\; \begin{cases}
\dfrac{y_1}{h_{m+1}(h_m + h_{m+1})} - \dfrac{y_1 - 2r}{2h_{m+1}}, & y_n = k_1, \\[3mm]
\dfrac{y_n}{h_{m+1}(h_m + h_{m+1})} + \dfrac{\rho\sigma y_n}{2h_{m+1} k_n} - \dfrac{y_n - 2r}{2h_{m+1}}, & k_1 < y \le 2r, \\[3mm]
\dfrac{y_n}{h_{m+1}(h_m + h_{m+1})} + \dfrac{\rho\sigma y_n}{2h_{m+1} k_n}, & 2r < y_n \le Y - k_{N+1},
\end{cases}
$$

where

$$
\alpha_{m,n} \;=\; -\dfrac{y_n}{h_m h_{m+1}} - \dfrac{\sigma^2 y_n}{k_{n+1}(k_n + k_{n+1})} - \dfrac{\rho\sigma y_n}{2h_m k_{n+1}},
$$

$$
\beta_{m,n} \;=\; -\dfrac{y_n}{h_m h_{m+1}} - \dfrac{\sigma^2 y_n}{k_n k_{n+1}} - \dfrac{\rho\sigma y_n}{2h_{m+1} k_n} - \dfrac{\rho\sigma y_n}{2h_m k_{n+1}},
$$

$$
\gamma_{m,n} \;=\; -\dfrac{y_n}{h_m h_{m+1}} - \dfrac{\sigma^2 y_n}{k_n(k_n + k_{n+1})} - \dfrac{\rho\sigma y_n}{2h_{m+1} k_n}.
$$

It is observed that, in the event $\rho = -1$, we have the following, due to (23):

$$h_{\min} = h_{\max} = h,\ k_{\min} = k_{\max} = k,\ k = \sigma h,$$

which indicates that uniform spatial grids must be employed. Thus, (25) reduces to

$$w'(\tau) = A_s w(\tau) + f_s(\tau).$$

Nontrivial entries of $A_s$ are readily obtained, based on the above discussion.

### 2.1.2. Case for $\rho \in (0, 1]$

We need the following restrictions on mesh steps, in the case [18]

$$\rho k_{\max} \leq \sigma h_{\min} \leq \sigma h_{\max} \leq \frac{1}{\rho} k_{\min}. \tag{26}$$

Apparently, when $\rho = 1$, the above implies that a uniform spatial mesh with $h = \sigma k$ must be used. Differing from (24), we consider a new, dynamically balanced cross-derivative approximation,

$$u_{xy}(x_m, y_n, \tau) = \frac{1}{2}(\Delta_{x,-}\Delta_{y,-} + \Delta_{x,+}\Delta_{y,+})u_{m,n}(\tau) + \mathcal{O}(h_{\max} + k_{\max}). \tag{27}$$

Computational stencils for (24) and (27) are shown in Figure 2.

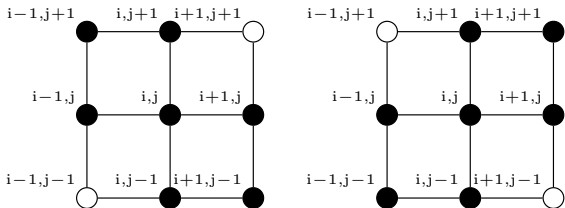

**Figure 2.** Computational stencils of (24) (**left**) and (27) (**right**).

In this circumstance, we obtain the following new system

$$w'(\tau) = \tilde{A}w(\tau) + \tilde{f}(\tau), \tag{28}$$

where $w, \tilde{f}(\tau) \in \mathbb{R}^{MN}$ and $\tilde{A} \in \mathbb{R}^{MN \times MN}$ is block tridiagonal; that is,

$$\tilde{A} = \begin{bmatrix} \tilde{D}_1 & \tilde{Q}_1 & \cdots & \cdots & \cdots & 0 \\ \tilde{P}_2 & \tilde{D}_2 & \tilde{Q}_2 & \cdots & \cdots & 0 \\ \vdots & \ddots & \ddots & \ddots & \cdots & \vdots \\ \vdots & \cdots & \tilde{P}_{M-2} & \tilde{D}_{M-2} & \tilde{Q}_{M-2} & 0 \\ \cdots & \cdots & \cdots & \tilde{P}_{M-1} & \tilde{D}_{M-1} & \tilde{Q}_{M-1} \\ 0 & \cdots & \cdots & \cdots & \tilde{P}_M & \tilde{D}_M \end{bmatrix}.$$

Nontrivial entries of $\tilde{P}_m$, $\tilde{D}_m$, and $\tilde{Q}_m$, within their respective ranges of $m$, are given by

$$\tilde{p}_{n,n-1}^{(m)} = \frac{\rho\sigma y_n}{2h_m k_n}, \quad y_n > k_1;$$

$$\tilde{p}_{n,n}^{(m)} = \begin{cases} \dfrac{y_1}{h_m(h_m + h_{m+1})}, & y_n = k_1, \\[2ex] \dfrac{y_n}{h_m(h_m + h_{m+1})} - \dfrac{\rho\sigma y_n}{2h_m k_n}, & k_1 < y_n \le 2r, \\[2ex] \dfrac{y_n}{h_m(h_m + h_{m+1})} - \dfrac{\rho\sigma y_n}{2h_m k_n} + \dfrac{y_n - 2r}{2h_m}, & 2r < y_n \le Y - k_{N+1}; \end{cases}$$

$$\tilde{r}_{n,n-1}^{(m)} = \begin{cases} \dfrac{\sigma^2 y_n}{k_n(k_n + k_{n+1})} - \dfrac{\rho\sigma y_n}{2h_m k_n}, & k_1 < y_n \le \eta, \\[2ex] \dfrac{\sigma^2 y_n}{k_n(k_n + k_{n+1})} - \dfrac{\rho\sigma y_n}{2h_m k_n} - \dfrac{\kappa(\eta - y_n)}{k_n}, & \eta < y_n \le Y - k_{N+1}; \end{cases}$$

$$\tilde{r}_{n,n}^{(m)} = \begin{cases} \tilde{\alpha}_{m,1} + \dfrac{y_1 - 2r}{2h_{m+1}} - \dfrac{\kappa(\eta - y_1)}{k_{n+1}}, & y_1 = k_1, \\[2ex] \tilde{\beta}_{m,n} + \dfrac{y_n - 2r}{2h_{m+1}} - \dfrac{\kappa(\eta - y_n)}{k_{n+1}}, & k_1 < y_n \le 2r, \\[2ex] \tilde{\beta}_{m,n} - \dfrac{y_n - 2r}{2h_m} - \dfrac{\kappa(\eta - y_n)}{k_{n+1}}, & 2r < y_n \le \eta, \\[2ex] \tilde{\beta}_{m,n} - \dfrac{y_n - 2r}{2h_m} + \dfrac{\kappa(\eta - y_n)}{k_n}, & \eta < y_n < Y - k_{N+1}, \\[2ex] \tilde{\gamma}_{m,N} - \dfrac{y_N - 2r}{2h_m} + \dfrac{\kappa(\eta - y_N)}{k_N}, & y_N = Y - k_{N+1}; \end{cases}$$

$$\tilde{r}_{n,n+1}^{(m)} = \begin{cases} \dfrac{\sigma^2 y_n}{k_{n+1}(k_n + k_{n+1})} - \dfrac{\rho\sigma y_n}{2h_{m+1}k_{n+1}} + \dfrac{\kappa(\eta - y_n)}{k_{n+1}}, & 0 < y_n \le \eta, \\[2ex] \dfrac{\sigma^2 y_n}{k_{n+1}(k_n + k_{n+1})} - \dfrac{\rho\sigma y_n}{2h_{m+1}k_{n+1}}, & \eta < y_n < Y - k_{N+1}; \end{cases}$$

$$\tilde{q}_{n,n}^{(m)} = \begin{cases} \dfrac{y_n}{h_{m+1}(h_m + h_{m+1})} - \dfrac{\rho\sigma y_n}{2h_{m+1}k_{n+1}} - \dfrac{y_n - 2r}{2h_{m+1}}, & 0 < y_n \le 2r, \\[2ex] \dfrac{y_n}{h_{m+1}(h_m + h_{m+1})} - \dfrac{\rho\sigma y_n}{2h_{m+1}k_{n+1}}, & 2r < y_n < Y - k_{N+1}, \\[2ex] \dfrac{y_N}{h_{m+1}(h_m + h_{m+1})}, & y_N = Y - k_{N+1}; \end{cases}$$

$$\tilde{q}_{n,n+1}^{(m)} = \frac{\rho\sigma y_n}{2h_{m+1}k_{n+1}}, \quad 0 < y_n < Y - k_{N+1},$$

where

$$\tilde{\alpha}_{m,n} = -\frac{y_n}{h_m h_{m+1}} - \frac{\sigma^2 y_n}{k_{n+1}(k_n + k_{n+1})} + \frac{\rho\sigma y_n}{2h_{m+1}k_{n+1}} + \frac{\rho\sigma y_n}{2h_m k_n},$$

$$\tilde{\beta}_{m,n} = -\frac{y_n}{h_m h_{m+1}} - \frac{\sigma^2 y_n}{k_n k_{n+1}} + \frac{\rho\sigma y_n}{2h_{m+1}k_{n+1}} + \frac{\rho\sigma y_n}{2h_m k_n},$$

$$\tilde{\gamma}_{m,n} = -\frac{y_n}{h_m h_{m+1}} - \frac{\sigma^2 y_n}{k_n(k_n + k_{n+1})} + \frac{\rho\sigma y_n}{2h_m k_n}.$$

The semi-discretized method (28) reduces to a uniform scheme when $\rho = 1$; that is,

$$w'(\tau) = \frac{1}{2h^2}\tilde{A}_s w(\tau) + \tilde{f}(\tau).$$

The nontrivial elements of $\tilde{A}$ can be determined from simplifications of the above formulae.

*2.2. Numerical Stability*

It is readily verified that the the solution to (25) is

$$w(\tau_{n+1}) = e^{\Delta\tau A}w(\tau_n) + \int_{\tau_n}^{\tau_{n+1}} e^{(t-\tau_n)A}f(t)dt, \quad n = 0, 1, \ldots, \tag{29}$$

where $\tau_n = n\Delta\tau$. The formal solution to (28) is similar. We have:

**Lemma 1** ([10,19]). *The semi-discretized schemes (25) and (28) are stable if*

$$\lim_{h_{max}, k_{max} \to 0} \left( \max_{\tau \in [0,\tau^*]} \left\| e^{\tau A} \right\|_2 \right) \leq c(\tau^*), \quad \lim_{h_{max}, k_{max} \to 0} \left( \max_{\tau \in [0,\tau^*]} \left\| e^{\tau \tilde{A}} \right\|_2 \right) \leq c(\tau^*),$$

*where $\tau^* \in (0, T)$.*

**Lemma 2** ([10]). *Let $B \in \mathbb{C}^{d \times d}$. Then $\sigma(B) \subset \cup_{i=1}^d S_i$, where*

$$S_i = \left\{ z \in C : |z - b_{i,i}| \leq \sum_{j=1, j \neq i}^d |b_{i,j}| \right\}$$

*are Geršhgorin discs, and $\sigma(B)$ is the set of all eigenvalues of B. Moreover, $\lambda \in \sigma(B)$ may lie on $\partial S_{i^0}$ for some $i^0 \in \{1, 2, \ldots, d\}$, only if $\lambda \in \partial S_i$ for all $i = 1, 2, \ldots, d$.*

**Lemma 3** ([20]). *The matrix exponential $e^{tA}$ tends to a zero matrix as $t \to +\infty$ if and only if all the eigenvalues of A have strictly negative real parts.*

**Theorem 1.** *The semi-discretized schemes (25) and (28) are linearly stable.*

**Proof.** We will only need to show the case of $\rho \in (0, 1]$, $\eta > 2r$ for (25), since extensions of our results for other cases are technically imminent. Thus, we only need to show that each of the *MN* Geršhgorin discs of *A* lies on the left side of the complex plane. In fact, there are five types of the Geršhgorin discs to consider:

1. Discs centered at an internal mesh point;
2. discs centered on one of the Dirichlet boundaries;
3. discs centered on the Neumann boundary;
4. discs centered at one of the intersection mesh points of two Dirichlet boundaries; and
5. discs centered at one of the intersection mesh points of one Dirichlet boundary and the Neumann boundary.

We provide detailed proofs for the first three types of discs. Similar arguments can be applied to the rest of the cases.

Case 1: In this situation, we first consider the situation in which $\eta < y_n \leq Y$. Let $z \in S_i$ be any complex number, where $S_i$ is a Geršhgorin disc centered at an internal point of the spatial grids. Thus,

$$\begin{aligned}
&\left| z + \frac{y_n}{h_m h_{m+1}} + \frac{\sigma^2 y_n}{k_n k_{n+1}} - \frac{\rho \sigma y_n}{2h_{m+1}k_{n+1}} - \frac{\rho \sigma y_n}{2h_m k_n} + \frac{y_n - 2r}{2h_m} - \frac{\kappa(\eta - y_n)}{k_n} \right| \\
&\leq \left| \frac{\sigma^2 y_n}{k_n(k_n + k_{n+1})} - \frac{\rho \sigma y_n}{2h_m k_n} - \frac{\kappa(\eta - y_n)}{k_n} \right| + \left| \frac{\sigma^2 y_n}{k_{n+1}(k_n + k_{n+1})} - \frac{\rho \sigma y_n}{2h_{m+1}k_{n+1}} \right| \\
&\quad + \left| \frac{y_n}{h_{m+1}(h_m + h_{m+1})} - \frac{\rho \sigma y_n}{2h_{m+1}k_{n+1}} \right| + \left| \frac{\rho \sigma y_n}{2h_{m+1}k_{n+1}} \right| + \left| \frac{\rho \sigma y_n}{2h_m k_n} \right| \\
&\quad + \left| \frac{y_n}{h_m(h_m + h_{m+1})} - \frac{\rho \sigma y_n}{2h_m k_n} + \frac{y_n - 2r}{2h_m} \right|.
\end{aligned} \tag{30}$$

Let $\alpha$ be the real part of $z$. Since we are concerned only about the upper bound of the real part of the eigenvalues, we may replace $z$ by $\alpha$ via a triangle inequality, and remove the absolute value sign on the left hand side of (30). As a consequence, (30) renders to

$$
\begin{aligned}
\alpha &+ \frac{y_n}{h_m h_{m+1}} + \frac{\sigma^2 y_n}{k_n k_{n+1}} - \frac{\rho \sigma y_n}{2 h_{m+1} k_{n+1}} - \frac{\rho \sigma y_n}{2 h_m k_n} + \frac{y_n - 2r}{2 h_m} - \frac{\kappa(\eta - y_n)}{k_n} \\
&\leq \left| \frac{\sigma^2 y_n}{k_n(k_n + k_{n+1})} - \frac{\rho \sigma y_n}{2 h_m k_n} - \frac{\kappa(\eta - y_n)}{k_n} \right| + \left| \frac{\sigma^2 y_n}{k_{n+1}(k_n + k_{n+1})} - \frac{\rho \sigma y_n}{2 h_{m+1} k_{n+1}} \right| \\
&+ \left| \frac{y_n}{h_{m+1}(h_m + h_{m+1})} - \frac{\rho \sigma y_n}{2 h_{m+1} k_{n+1}} \right| + \left| \frac{\rho \sigma y_n}{2 h_{m+1} k_{n+1}} \right| + \left| \frac{\rho \sigma y_n}{2 h_m k_n} \right| \\
&+ \left| \frac{y_n}{h_m(h_m + h_{m+1})} - \frac{\rho \sigma y_n}{2 h_m k_n} + \frac{y_n - 2r}{2 h_m} \right|.
\end{aligned}
\tag{31}
$$

Recall (26) and that $\rho > 0$. We have

$$
\frac{2}{\rho \sigma} k_n, \ \frac{2}{\rho \sigma} k_{n+1} \geq h_m + h_{m+1} \text{ and } h_m, \ h_{m+1} \geq \frac{\rho}{\sigma}(k_n + k_{n+1}).
$$

The above indicates that

$$
\begin{aligned}
\frac{\sigma^2 y_n}{k_n(k_n + k_{n+1})} &\geq \frac{\rho \sigma y_n}{2 h_m k_n}, \\
\frac{\sigma^2 y_n}{k_{n+1}(k_n + k_{n+1})} &\geq \frac{\rho \sigma y_n}{2 h_{m+1} k_{n+1}}, \\
\frac{y_n}{h_{m+1}(h_m + h_{m+1})} &\geq \frac{\rho \sigma y_n}{2 h_{m+1} k_{n+1}}, \\
\frac{y_n}{h_m(h_m + h_{m+1})} &\geq \frac{\rho \sigma y_n}{2 h_m k_n}.
\end{aligned}
$$

Furthermore, since $y > \eta > 2r$, we conclude that

$$
-\frac{\kappa(\eta - y_n)}{k_n} \geq 0 \text{ and } \frac{y_n - 2r}{2 h_m} \geq 0.
$$

Therefore, the term inside each pair of absolute signs in (31) must be positive. We may remove all absolute signs in (31), and subsequently yield

$$
\alpha \leq 0,
$$

which is what we expect. Generalizations of the discussion for cases involving $y \leq \eta$ are straightforward. Therefore, all eigenvalues contained in $S_i$ must lie on the left half of the complex plane.

Case 2: Without loss of generality, we consider the case where $x = x_1$ and $\eta < y < Y$. Thus, for any complex number $z \in S_i$, where $S_i$ is a Geršhgorin disc satisfying

$$
\begin{aligned}
\left| z \right. &+ \frac{y_n}{h_m h_{m+1}} + \frac{\sigma^2 y_n}{k_n k_{n+1}} - \frac{\rho \sigma y_n}{2 h_{m+1} k_{n+1}} - \frac{\rho \sigma y_n}{2 h_m k_n} + \frac{y_n - 2r}{2 h_m} - \left. \frac{\kappa(\eta - y_n)}{k_n} \right| \\
&\leq \left| \frac{\sigma^2 y_n}{k_n(k_n + k_{n+1})} - \frac{\rho \sigma y_n}{2 h_m k_n} - \frac{\kappa(\eta - y_n)}{k_n} \right| + \left| \frac{\sigma^2 y_n}{k_{n+1}(k_n + k_{n+1})} - \frac{\rho \sigma y_n}{2 h_{m+1} k_{n+1}} \right| \\
&+ \left| \frac{y_n}{h_{m+1}(h_m + h_{m+1})} - \frac{\rho \sigma y_n}{2 h_{m+1} k_{n+1}} \right| + \left| \frac{\rho \sigma y_n}{2 h_{m+1} k_{n+1}} \right|.
\end{aligned}
$$

Similar to the previous case, we take $\alpha$ to be the real part of $z$. Thus,

$$
\alpha \leq \frac{y_n}{h_{m+1}} \left( \frac{1}{h_m + h_{m+1}} - \frac{1}{h_m} \right) - \frac{y_n - 2r}{2 h_m} < 0.
$$

The above implies that such an $S_i$ must lie strictly on the left half of the complex plane, and the origin cannot be on its boundary. This ensures our expectation.

Case 3: In this circumstance, the Geršhgorin discs $S_i$ concerned are centered at boundary points where a Neumann condition is imposed. Hence, for any $z \in S_i$ we have

$$
\left| z + \frac{y_N}{h_m h_{m+1}} + \frac{\sigma^2 y_N}{k_N(k_{N+1} + k_N)} - \frac{\rho \sigma y_N}{2h_m k_N} + \frac{y_N - 2r}{2h_m} - \frac{\kappa(\eta - y_N)}{k_N} \right|
$$
$$
\leq \left| \frac{\sigma^2 y_N}{k_N(k_N + k_{N+1})} - \frac{\rho \sigma y_N}{2h_m k_N} - \frac{\kappa(\eta - y_N)}{k_N} \right| + \left| \frac{y_N}{h_{m+1}(h_m + h_{m+1})} \right|
$$
$$
+ \left| \frac{\rho \sigma y_N}{2h_m k_N} \right| + \left| \frac{y_N}{h_m(h_m + h_{m+1})} - \frac{\rho \sigma y_N}{2h_m k_N} + \frac{y_N - 2r}{2h_m} \right|.
$$

The above indicates that $\alpha$, the real part of $z$, must satisfy

$$
\alpha \leq \frac{y_N}{(h_m + h_{m+1})^2} - \frac{y_N}{h_m h_{m+1}} < 0.
$$

Recall Lemma 2. Since the origin cannot lie on the boundary of every Geršhgorin disc, combining results from the three cases, we conclude immediately that all eigenvalues of $A$ must be strictly in the left half of the complex plane. Thus, we must have

$$
\lim_{h_{max}, k_{max} \to 0} \left( \max_{\tau \in [0, \tau^*]} \left\| e^{\tau A} \right\|_2 \right) \leq c(\tau^*).
$$

The above completes our proof. $\square$

*2.3. Computational Experiments*

Consider (9)–(14). Similar to the discussions by Zhu et al. [14], we fix $X = 8$, $Y = 1$. We first concentrate on experiments with $\rho = -0.5$ and $T = 0.5$. Next, to test against extreme cases in the option market, we proceed with $\rho = -1$ and $T = 5$. For demonstrating the numerical solution and its rate of convergence estimates, we first consider uniform spatial grids. To this end, we denote

$$
h_m = h, \ k_n = k = \sigma h, \ m = 1, 2, \ldots, M; \ n = 1, 2, \ldots, N.
$$

Results over nonuniform grids will be presented later.

Some key parameters used are shown in Table 1. Further, a Crank-Nicolson type temporal integrator will be utilized for advancing our semi-discretized system (25), (28), with $\Delta \tau$ as the temporal step [19]. It is known that $\lambda = \Delta \tau / c^2$, where $c = \min \{h, k\}$, plays the effective role of the Courant number [21,22]. We experiment with different values of $\lambda$, varying from 0.5 to 1.

**Table 1.** Key parameter values for numerical simulations.

| Key Parameter | Value Used |
|---|---|
| Strike price | $K = 100$ |
| Volatility of volatility | $\sigma = 1$ |
| Risk-free interest rate | $r = 0.05$ |
| Mean reversion speed | $\kappa = 2$ |
| Long-run mean of volatility | $\eta = 0.1$ |

Our semi-discretized scheme is expected to be up to the first order in convergence in space. To numerically examine this by experiment, we employ a generalized Milne's device [10,18]. Then, for a selected terminal time $T$, we denote the numerical solution at point $(x_m, y_n, T)$, $1 \leq m \leq M, 1 \leq n \leq N$, as $u_{m,n;h}$ for any particular spatial step $0 < h \ll 1$. Likewise, we let

$u_{m,n;h/2}$ and $u_{m,n;h/4}$ be the computed solutions obtained by using $h/2$ and $h/4$, respectively. Thus, the point-wise rate of spatial convergence at $T$ can be evaluated via

$$R^h_{m,n} \approx \frac{1}{\ln 2} \ln \frac{\left| u_{m,n;h} - u_{m,n;h/2} \right|}{\left| u_{m,n;h/2} - u_{m,n;h/4} \right|}. \tag{32}$$

Most of our experiments are accomplished on Apple workstations. MATLAB platforms without parallelizations are used throughout our operations.

Let $h = 0.01$ and $\sigma = 1$. For simplicity of notation, we use the same letter $v$ for the approximate solution to (4). We show the solution $v$ for $\rho = -0.5$ and $\rho = -1$ in Figures 3 and 4, respectively. To see more precise solution profiles, we show corresponding contour maps next to the surfaces. It can be observed that the European put option price is a decreasing function of the stock price $S$. This coincides well with the financial theory that a put option price should have a negative correlation with the underline stock price [11,23]. To examine further the delicate relationship between a put option price and its volatility, we plot an average numerical solution $\bar{v}(y,t)$ taken across different stock prices with respect to the volatility in Figure 5. The simulated computational result is exactly what we would expect, since a put option price should be positively correlated with volatility [22,23].

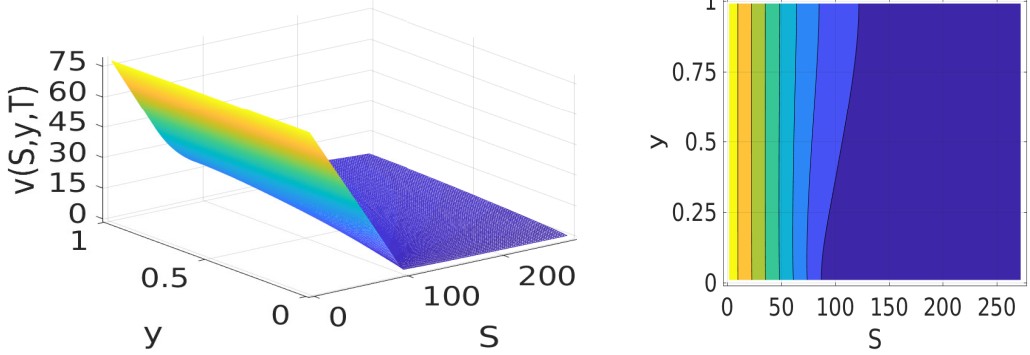

**Figure 3.** (**Left**) Price of an European put option at $T = 0.5$ and for $\rho = -0.5$; (**Right**) Corresponding contour map.

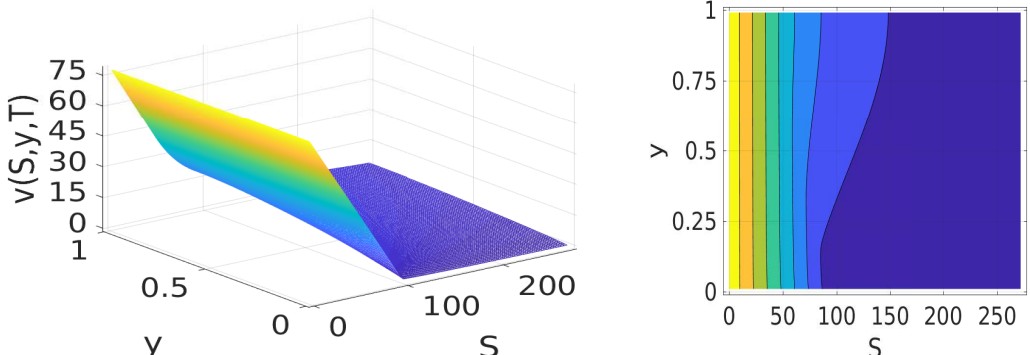

**Figure 4.** (**Left**) Price of an European put option at $T = 5$ and for $\rho = -1$; (**Right**) Corresponding contour map.

We plot the computed rate of convergence surfaces for the cases $\rho = -0.5$ and $\rho = -1$ in Figures 6 and 7, respectively. In addition, a summary of point-wise convergence rates for the circumstance as $\rho = -0.5$, $T = 0.5$ on different spatial grids is given in Table 2. Minor disturbances can be observed in regions where the solution changes fast, particularly in the extreme situations with $\rho = -1$, as demonstrated in Figure 7. These results are consistent with those from well-established high-order schemes [2,4,8,13,14]. However, due to the low-order nature of our scheme, to get the same accuracy we have to employ relatively small mesh sizes. This results in longer running

times for our scheme. But, because of the established theoretical results, the new method ensures excellent stability and a great structure for accommodating an exponential splitting, which will improve computational efficiency in higher dimensions. Improving computational efficiency through exponential splitting methods, particularly variable step ADI or LOD approximations [12,18,24], is one of our ongoing researches.

**Table 2.** Rates of convergence $R_{PW}^h$ observed with $\sigma = 1$, $\rho = -0.5$, and $T = 0.5$.

| Mesh Steps | Rconv. Rates | $\lambda = 0.5$ | $\lambda = 0.75$ | $\lambda = 1$ |
|---|---|---|---|---|
| | $\min_{m,n}(R_{m,n}^h)$ | 0.6193 | 0.6134 | 0.6026 |
| $h = 0.01$ | $\max_{m,n}(R_{m,n}^h)$ | 1.0024 | 0.9976 | 0.9811 |
| | $\text{mean}_{m,n}(R_{m,n}^h)$ | 0.9026 | 0.90438 | 0.9053 |
| | $\min_{m,n}(R_{m,n}^h)$ | 0.6324 | 0.6221 | 0.6206 |
| $h = 0.02$ | $\max_{m,n}(R_{m,n}^h)$ | 0.9674 | 1.0007 | 1.0151 |
| | $\text{mean}_{m,n}(R_{m,n}^h)$ | 0.8342 | 0.8300 | 0.8296 |
| | $\min_{m,n}(R_{m,n}^h)$ | 0.5824 | 0.5971 | 0.6179 |
| $h = 0.03$ | $\max_{m,n}(R_{m,n}^h)$ | 0.9941 | 0.9437 | 0.9586 |
| | $\text{mean}_{m,n}(R_{m,n}^h)$ | 0.7952 | 0.8015 | 0.8142 |

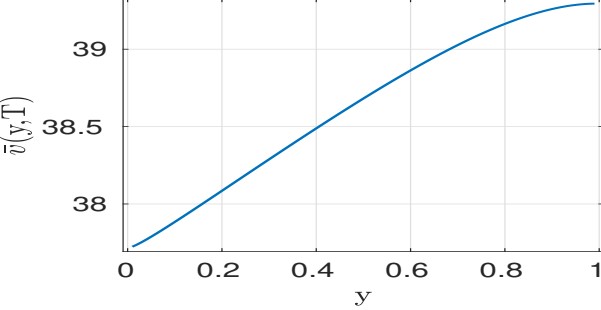

**Figure 5.** Relationship between the average price of an European put option with volatility $T = 0.5$ and $\rho = -0.5$.

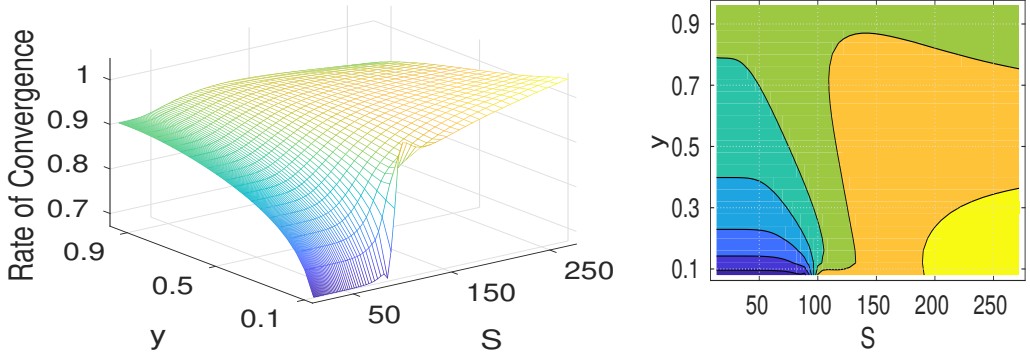

**Figure 6.** (**Left**) Pointwise rate of convergence estimate, $T = 0.5$ and $\rho = -0.5$; (**Right**) Corresponding contour map.

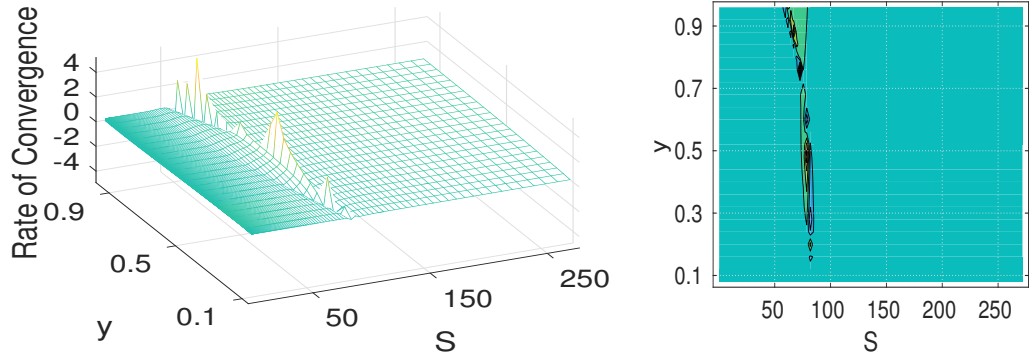

**Figure 7.** (**Left**) Point-wise rate of convergence estimate, $T = 5$ and $\rho = -1$. (**Right**) Corresponding contour map.

Now, we consider simulations over nonuniform spatial grids. To better design our tests, we are particularly interested in the following nonlinear distribution governing functions

$$z_1(S) = \sqrt{\frac{1}{2.56} + \frac{25(S/K)^{10}}{2.56[1 + (S/K)^5]^4}}, \quad S_{\min} \leq S \leq S_{\max}, \tag{33}$$

$$z_2(y) = \frac{10\sqrt{0.5y}}{7}, \quad y_{\min} \leq y \leq y_{\max}. \tag{34}$$

In our simulation experiments, selections of monitoring functions are based initially on the numerical solution $v$, acquired on uniform spatial meshes. The monitoring function in the $S$-direction, $z_1$, is chosen so more grid points will be distributed around the areas where oscillatory convergence rates appear, as indicated in Figure 7. On the other hand, the monitoring function in the $y$-direction, $z_2$, is chosen in such a way that it matches the trend of solution curvature in direction, as shown in Figure 5. In this way, more mesh points can be relocated ideally to regions where the solution has sharper increases. An overall more accurate result is thus anticipated [19]. For detailed information on general mesh adaptations, we refer the reader to Cheng et al. [24] and Sheng and Padgett [18]. Our nonuniform grids are generated via an arc-length equal-distribution principal for the functions $z_1, z_2$ in the $S$- and $y$-directions, respectively. The principal is commonly utilized in adaptive computations, and serves as an initial exploration for more sophisticated adaptations [18,24]. The calculation of the mesh coordinates in our experiments is conducted based on a forward Euler formula for arc-lengths [19]. For instance, in the $S$-direction we have

$$S_{j+1} = S_j + \frac{\ell}{(N-1)\sqrt{1 + [(z_1(S_j))_S]^2}}, \quad j = 1, 2, \ldots N - 1, \tag{35}$$

where $\ell$ is the total arc-length; that is,

$$\ell = \int_{S_{\min}}^{S_{\max}} \sqrt{1 + [(z_1(S))_S]^2} \, dS. \tag{36}$$

While the distribution functions $z_1, z_2$ are shown in Figure 8, their composite surface plots can be found in Figure 9. The latter characterizes the 2-dimensional profile of our grid's distribution. The numerical solution acquired over such nonuniform grids, with $\rho = -0.5$ and $T = 0.5$, is given in Figure 10.

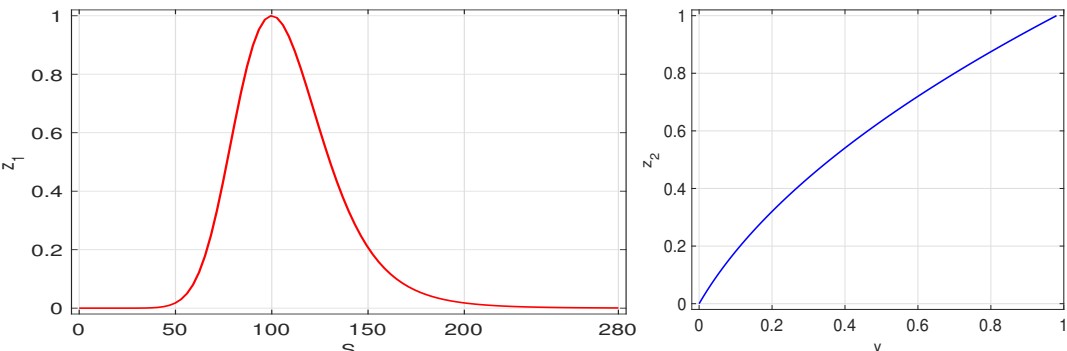

**Figure 8.** (**Left**) Nonlinear grid distribution governing the function $z_1$ in the $S$-direction; (**Right**) Nonlinear mesh distribution governing the function $z_2$ in the $y$-direction.

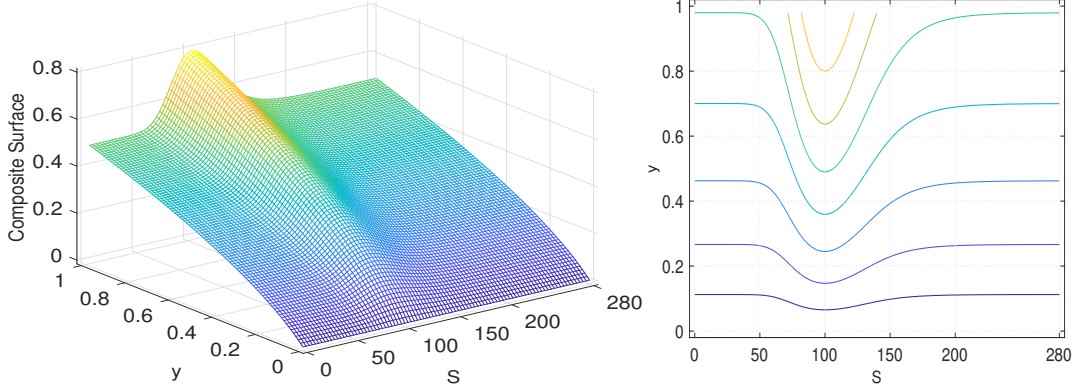

**Figure 9.** (**Left**) A composite surface plot of $z_1(S)z_2(y)$; (**Right**) Corresponding contour map.

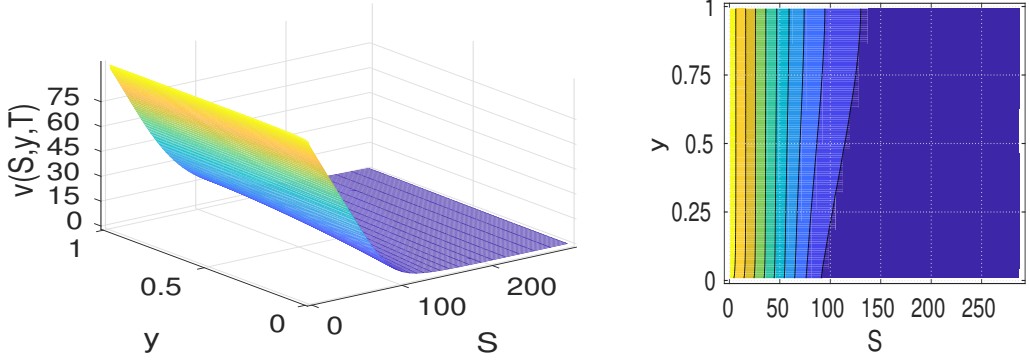

**Figure 10.** (**Left**) Price of an European put option on nonuniform grids, $T = 0.5$ and $\rho = -0.5$; (**Right**) Corresponding contour map.

Let $\Omega_{N,M}$ be a reference spatial mesh, which can be either our uniform or nonuniform mesh. We may map solutions $v_{\text{unif}}$ and $v_{\text{nonunif}}$, numerical solutions obtained on the uniform mesh and nonuniform mesh, respectively, to $\Omega_{N,M}$. We then define the following point-wise relative difference function, $R_d$.

$$R_d(S, y, t) = \frac{|v_{\text{unif}}(S, y, t) - v_{\text{nonunif}}(S, y, t)|}{|v_{\text{unif}}(S, y, t)|}, \quad (S, y, t) \in \Omega_{N,M}, \ 0 < t \leq T. \tag{37}$$

In our numerical approaches, we let the uniform mesh be our reference spatial mesh and then map $v_{\text{nonunif}}$ to $\Omega_{N,M}$, utilizing the built-in MATLAB subroutine interp2.m.

The interesting surface of the point-wise difference and its contour plot are given in Figure 11.

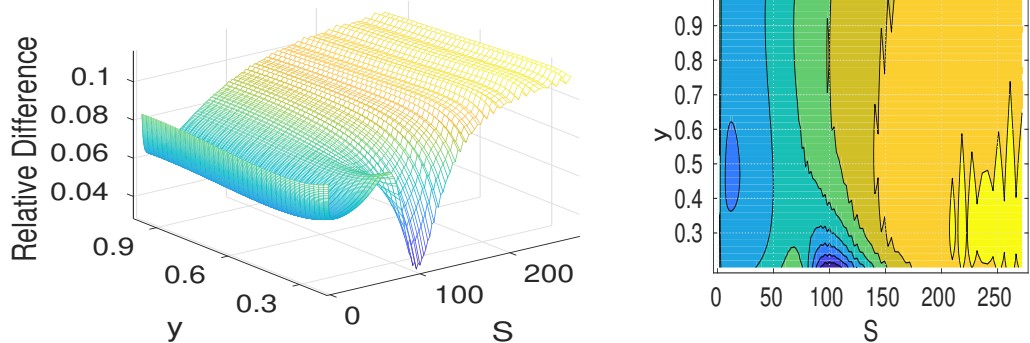

**Figure 11.** (**Left**) Relative difference between solutions on uniform and nonuniform grids, $T = 0.5$ and $\rho = -0.5$; (**Right**) Corresponding contour map. Formulas (9) and (10) are used.

We may further calculate the relative difference index $\delta$ using the standard 2-norm; that is,

$$\delta_2 = \frac{\|v_{\text{unif}} - v_{\text{nonunif}}\|_2}{\|v_{\text{unif}}\|_2}. \tag{38}$$

For numerical solutions and the difference function illustrated in Figures 3 and 10, we have $\delta_2 \approx 0.067$.

## 3. Discussion

A dynamically balanced up-downwind semi-discretized finite difference method was constructed and analyzed in this paper, based on arbitrary spatial grids. The algorithm acquired was easy to use. It was also effective for solving the underlying Heston stochastic volatility option-pricing model problems with cross-derivative terms. The scheme was proven to be numerically stable. Computational experiments were carried out to verify our expectations, both on uniform and nonuniform grids. The numerical method is expected to be first order in space.

The spectral norm was used throughout this paper. The study can be extended by using different Euclidean norms. Our ongoing research is in including effective schemes on variable spatial and temporal meshes for different financial products and simulations. We have also been considering effective adaptation strategies, such as those investigated by Meng and Padgett et al. [19,22].

Our future endeavors also include improving computational efficiency through exponential splitting methods, particularly variable-step ADI or LOD approximations [12,18,24]. Compact schemes for raising accuracy have also been introduced in our study, with initial successes in handling cross-derivatives dynamically, and well balances pricing American and some Asian options [9,11,14,16]. Initial investigations were very promising.

## 4. Materials and Methods

Most parts of our computations are carried out on a MATLAB® platform and its parallel computing toolbox, on a high performance HP® C3000BL HPC cluster running CentOS® V, at Baylor University. The processor consists of 128 computer nodes, each with 32 GB of RAM and dual quad-core Intel 2.6 GHz processors, giving a total of 1024 cores. An Infiniband ConnectX® DDR network is used for message passing and networked storage. Shared storage capacity in the cluster is 123 TB. All computer programs are available, by request to the corresponding author.

**Author Contributions:** C.S. and Q.S. contributed equally to this work. Both authors have read and approved the final manuscript.

**Funding:** This research received no external funding.

**Acknowledgments:** The authors would like to thank the anonymous referees for their time and constructive comments. Their suggestions have not only elevated the quality and presentation of this paper, but also thrown a light on important extensions of this research. The authors also appreciate their colleagues, in particular Joshua Padgett and Tiffany Jones, for many extremely meaningful discussions.

**Conflicts of Interest:** The authors declare no conflict of interest.

## Abbreviations

The following abbreviations are used in this manuscript:

| | |
|---|---|
| ADI | Alternating direction implicit |
| BSM | Black-Scholes-Merton |
| DDR | Double data rate |
| HPC | High-performance computing |
| LOD | Local one-dimensional |
| RAM | Random access memory |
| GB | Gigabyte |
| TB | Terabyte |

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
