# Peer review of "An Exploration of a Balanced Up-Downwind Scheme for Solving Heston Volatility Model Equations on Variable Grids"

_algorithms, doi:10.3390/a12020030_

Round 1

Reviewer 1 Report

The authors clearly know what they do with respect to finite difference methods to solve partial differential equations. From this point of view the manuscript is well written and the mathematical equations are presented professionally. However there are a few shortcomings in the financial part that should be corrected before proceeding to publication.

1. The vast literature on the Heston model after the seminal paper of 1993 [9] is underreferenced. A recent paper which in its introduction gives a good overview was published by Cui et al. in Eur. J. Oper. Res. 263, 625 (2017). At the very least, the authors should add a citation to the latter between lines 20 and 24. After reading it, they may optionally expand slightly the oversimplified account they give in these 5 lines, possibly including a few further key citations from the bibliography of Cui et al., where I most notably suggest the monography by Rouah (2013), a standard book on the Heston model.

2. The practical relevance of the authors' contribution is not clear. The Heston model is popular because an analytic expression of its characteristic function is known and the price of European options can be computed straightforwardly from the latter without solving a PDE. Of course this may still be published as an interesting academic exercise to obtain the price in another way, but if there is more about it, e.g. for the pricing of exotic derivatives, it should be explained.

3. Although the authors do numerical experiments, they do not provide a solid validation of their results against established methods, and Table I gives only 4 "key" parameters of the simulation, whereas there are 5 model parameters and 3 or 4 market parameters for a total of 8 or 9, depending on whether a dividend yield or foreign interest rate is considered too.

Author Response

First of all, we would like to thank you  for your extremely valuable comments and suggestions.

Based on them, we have made various modifications in the revised version of our manuscript. All such changes are marked in red in the paper. The following attachment contains our responses to the your specific questions and comments.

Sincerely yours,

Chong Sun
(on behalf of Qin Sheng)

Reviewer 2 Report

In this article, the authors developed a numerical scheme to solve the Heston stochastic volatility option-pricing model. The model is based on a partial differential equation with two space variables: asset price and volatility. The numerical scheme includes approximation for the cross-derivatives. The method is first order and authors implemented in uniform and non-uniform grids. The authors prove the numerical stability and some numerical simulations are presented.

The article is interesting for the scientific community. Few things can be improved.

1-    In the introduction, the authors mentioned that the motivation is to propose a second order scheme and prove the numerical stability. However, in the Reference 5 the authors presented a higher order scheme and they study the numerical stability. The authors need to explain a little better the motivation taking this fact into account.

2-    In Figures 1 and 2, the authors present the stencils of the numerical scheme. The authors need to add the space directions on the stencil and the levels, i.e. i, i+1, j, j+1, so readers can understand easy the stencils.

3-     For the non-uniform meshes, I suggest to add one numerical example to extend the results for the non-uniform meshes.

4-    The authors chose the nonlinear distributions of the meshes using some particular functions. The rationale behind the particular functions is that they asymptotically fit into profiles of the option solution v as shown in experiments. The authors need to explain better this. For instance, a priori the solution is not known so how can we choose the mesh distribution a priori?

5-    The authors need to add some comments regarding why to use non-uniform meshes. Usually is to reduce computation time.

Line 146: “To exam actual performances”   please change it.

Author Response

First of all, we would like to thank you for your extremely valuable comments and suggestions. Based on them, we have made various modifications in the revised version of our manuscript. All such changes are marked in red in the paper. The following attachment contains our responses to your specific questions and comments.

Sincerely yours,

Chong Sun
(on behalf of Qin Sheng)

Round 2

Reviewer 1 Report

The authors have improved the manuscript in certain aspects, but because of too much hurry they have unfortunately made it worse in other aspects, so that a second revision is required.

1. The authors were pointed to two extra references plus optionally some citations therein contained. They got just one right (Rouah 2013) and quite incredibly screwed up the other (Cui 2017): I explicitly suggested Cui et al., Eur. J. Oper. Res. 263, 625 (2017), which is specifically on the Heston model, but they cited Cui et al., Eur. J. Oper. Res. 2017, 262, 381, which is by a different author (not Y. Cui but Z. Cui) on a remotely related subject (the title is "A general framework for discretely sampled realized variance derivatives in stochastic volatility models with jumps") without realizing they were on the wrong track because they clearly did not bother to read it, otherwise they would have noticed e.g. that the reference they erroneously picked (I guess they just searched for Cui Eur. J. Oper. Res. 2017 and simply took the top hit wihtout any further thought) does not cite Rouah although I said that the latter is the most notable item in the bibliography of Y. Cui et al. (2017). Therefore please substitute references 2, 5 and 6 with https://doi.org/10.1016/j.ejor.2017.05.018, whose title is "Full and fast calibration of the Heston stochastic volatility model". Its contribution goes much beyond calibration as it proposes a new and mathematically more convenient expression of the characteristic function of the Heston model, which is needed for pricing and calibration; it also discusses a comprehensive bibliography of the Heston model and explains the industry practice it is used for. The hurried and sloppy extension of the bibliography of this manuscript is shown also by the fact that in their answer the authors say that they added three references, while they actually added four.

2. The authors have made the practical relevance of their contribution clearer by mentioning American options. However, in this paper they do not price an American option. I do not ask that they deal with American options here, but they should discuss more clearly the rationale of presenting this as a preliminary exercise on the road to solve something actually useful. Moreover, I do not agree with the new sentences at the end of page 1 and the top of page 2 which state that the recent financial crisis has increased volatility risk, has caused more trading in volatility derivatives, and thus has attracted more research on stochastic volatility models. Volatility trading (options on the VIX) has been introduced at the CBOE much before the 2008 crisis; research on stochastic volatility in general and the Heston model in particular has been strong already before the crisis, as can be seen in the bibliography of Y. Cui et al. (2017) and Rouah (2013). The 2008 crisis has depressed volatility rather than increasing it; this was particularly evident in the FX, where interest rates differentials became negligible because most major central banks lowered their interest rates to close to zero.

3. In Table I the authors now specify a "volatility parameter" sigma which actually should be called "volatility of volatility", but they should provide also the initial value of the squared volatility (the Heston model has 5 parameters). They still mix model parameters with market parameters, and among the latter they still provide only K while they should provide at least S_0 and r too. The authos say they do not wish to complicate the picture specifying q (i.e. the foreign interest rate), but they should at least provide the domestic one, i.e. r. The authors still do not provide a solid validation nor benchmark of their numerical results against established methods; they just added the generic sentence "These results are consistent with those from well-established high-order schemes [7,8,13,18,26]." Please be more detailed.

4. In one of the new sentences, "we refer the reader to [4,24]" is not good referencing style. Please write "we refer to Cheng et al. [4] and to Sheng and Padgett [24]" or "we refer to Refs. 4 and 24".

Author Response

Thank you very much for your reviews concerning our manuscript entitled ``An exploration of a mixed up-downwind scheme for solving Heston volatility model equations on variable grids''.

We have now carefully completed our corrections and improvements in light of your comments and reviews. We appreciate very much for your time spent and comments shared. 

The detailed replies are in the attached pdf file. 

Thank your very much for your time!

Sincerely yours,

Chong Sun

(On behalf of Qin Sheng)

Reviewer 2 Report

The authors improved  the clarity of the paper.  One thing that still can be improved is regarding the comparison with previous higher order schemes.

The only comment is: "These results are consistent with those from well-established high-order schemes [7,8,13,18,26]."

Something additional should be added. For instance our method is accurate ? or faster ?

Author Response

Thank you very much for your kind letter and report concerning our manuscript entitled ``An exploration of a mixed up-downwind scheme  for solving Heston volatility model equations on variable grids''. We have now carefully completed our corrections and improvements in  light of your comments and reviews. We appreciate very much for your time spent and comments shared.  The detailed replies are in the attached pdf file.  Again, thank you very much for your time! Sincerely yours, Chong Sun (On behalf of Qin Sheng)

Round 3

Reviewer 1 Report

The manuscript has been improved enough to be accepted. A few comments for

optional revisions follow. A further review is not required.

1. The authors have swapped the three incorrectly picked references with the

one I originally suggested. This is now ok. Nevertheless, the discussion of the

Heston model and the successive literature about it is still basic. You may

expand this in your follow-up article. Minor corrections:

Line 27: additional sources -> an additional source

Lines 27-28: Heston proposed a different approach by introducing the

consideration of stochastic volatility -> Heston proposed a more refined

approach based on the concept of stochastic volatility

2. This is now better, although of course the new lines 33-38 are still

sketchy. Minor correction:

Line 32: initial-boundary conditions -> initial boundary conditions

3. Also this has improved, although the new lines 169-176 would of course

benefit from an expansion in order to report detailed numerical results about

the trade-off between accuracy and stability. Moreover the explanations on the

choice of the intervals for y and S should be moved from the answer letter to

the paper. A few further misunderstandings should be corrected: the volatility

is the square root of y(t), not y(t), which is the variance of S(t); S_0 = S(0)

is commonly used not only as the initial spot price, but also as a scale factor

to define the log price x(t) = s(t) = log(S(t)/S_0)) (in general, S is

a dimensional quantity expressed e.g. in dollars, so that one cannot take

the logarithm of it without dividing by another quantity like S_0 with the

same dimension) and more in general reduced units as one prefers to use in

numerical simulations. Failing stubbornly to define S_0 leads to introduce it

surrectitiously as the magic number 100 in the unnumbered equation for z_1(S)

after line 178 (the most common choices for S_0 are 1 and 100). Please number

all equations for reference, as you number all pages, sections, bibliographic

entries, figures and tables.

Minor corrections:

Table 1: as there is no foreign interest rate, change domestic interest rate to risk-free interest rate.

Line 169: low order nature -> low-order nature

Line 170: to get same accuracy with the established schemes -> to get the same accuracy

Line 171: longer running time -> longer running times

4. The authors have corrected the wrong referencing style in the example that

I made on page 17, where they now write "we refer to Cheng et al. [3] and to

Sheng and Padgett [22]". However, they have failed to realise that there

are other examples of the same wrong referencing style, e.g. in line 139

("Similar to discussions in [24]"), line 199 ("as those investigated in [17,

19]), and possibly elsewhere.

However, the scale factor (used e.g. in the equation for z_1(S) after line 178)
can also be set equal to the strike price K = 100.

Author Response

Thank you very much for your kind letter and report concerning our manuscript
entitled ``An exploration of a mixed up-downwind scheme for solving Heston volatility model equations on variable grids''.

We have now carefully completed our corrections and improvements in light of comments and

reviews from you. We appreciate very much  for your time spent and comments shared. 

The detailed reply was in the attachment and we have highlighted the revised parts by red in the new manuscript.

Thank you again for your time!

Sincerely,

Yours,

Chong Sun

(On behalf of Qin Sheng)
